# Assessing Anti-Social and Aggressive Behavior in a Zebrafish (*Danio rerio*) Model of Parkinson’s Disease Chronically Exposed to Rotenone

**DOI:** 10.3390/brainsci12070898

**Published:** 2022-07-08

**Authors:** Ovidiu-Dumitru Ilie, Raluca Duta, Roxana Jijie, Ilinca-Bianca Nita, Mircea Nicoara, Caterina Faggio, Romeo Dobrin, Ioannis Mavroudis, Alin Ciobica, Bogdan Doroftei

**Affiliations:** 1Department of Biology, Faculty of Biology, “Alexandru Ioan Cuza” University, Carol I Avenue, No 20A, 700505 Iasi, Romania; ovidiuilie90@yahoo.com (O.-D.I.); duta.raluca112@gmail.com (R.D.); mirmag@uaic.ro (M.N.); 2Department of Exact and Natural Sciences, Institute of Interdisciplinary Research, “Alexandru Ioan Cuza” University, Carol I Avenue, No 11, 700506 Iasi, Romania; 3Faculty of Medicine, University of Medicine and Pharmacy “Grigore T. Popa”, University Street, No 16, 700115 Iasi, Romania; ilinca.bi@yahoo.com (I.-B.N.); bogdandoroftei@gmail.com (B.D.); 4Doctoral School of Geosciences, Faculty of Geography-Geology, “Alexandru Ioan Cuza” University, Carol I Avenue, No 20A, 700505 Iasi, Romania; 5Department of Chemical, Biological, Pharmaceutical and Environmental Sciences, University of Messina, Viale F. Stagno d’Alcontre 31, 98166 Messina, Italy; caterina.faggio@unime.it; 6Department of Psychiatry, University of Medicine and Pharmacy “Grigore T. Popa”, University Street, No 16, 700115 Iasi, Romania; romeodobrin2002@gmail.com; 7Department of Neuroscience, Leeds Teaching Hospitals, NHS Trust, Leeds LS2 9JT, UK; i.mavroudis@nhs.net

**Keywords:** anti-social, aggressivity, counterclockwise rotation, zebrafish, *Danio rerio*, Parkinson’s disease, rotenone

## Abstract

Background: Rotenone (ROT) is currently being used in various research fields, especially neuroscience. Separated from other neurotoxins, ROT induces a Parkinson’s disease (PD)-related phenotype that mimics the associated clinical spectrum by directly entering the central nervous system (CNS). It easily crosses through the blood–brain barrier (BBB) and accumulates in mitochondria. Unfortunately, most of the existing data focus on locomotion. This is why the present study aimed to bring novel evidence on how ROT alone or in combination with different potential ant(agonists) might influence the social and aggressive behavior using the counterclockwise rotation as a neurological pointer. Material and Methods: Thus, we exposed zebrafish to ROT—2.5 µg/L, valproic acid (VPA)—0.5 mg/mL, anti-parkinsonian drugs (LEV/CARB)—250 mg + 25 mg, and probiotics (PROBIO)—3 g for 32 days by assessing the anti-social profile and mirror tests and counterclockwise rotation every 4 days to avoid chronic stress. Results: We observed an abnormal pattern in the counterclockwise rotation only in the (a) CONTROL, (c) LEV/CARB, and (d) PROBIO groups, from both the top and side views, this indicating a reaction to medication and supplements administered or a normal intrinsic feature due to high levels of stress/anxiety (*p* < 0.05). Four out of eight studied groups—(b) VPA, (c) LEV/CARB, (e) ROT, and (f) ROT + VPA—displayed an impaired, often antithetical behavior demonstrated by long periods of time on distinct days spent on the right and the central arm (*p* < 0.05, 0.005, and 0.0005). Interestingly, groups (d) PROBIO, (g) ROT + LEV/CARB, and (h) ROT + PROBIO registered fluctuations but not significant ones in contrast with the above groups (*p* > 0.05). Except for groups (a) CONTROL and (d) PROBIO, where a normalized trend in terms of behavior was noted, the rest of the experimental groups exhibited exacerbated levels of aggression (*p* < 0.05, 0.005, and 0.001) not only near the mirror but as an overall reaction (*p* < 0.05, 0.005, and 0.001). Conclusions: The (d) PROBIO group showed a significant improvement compared with (b) VPA, (c) LEV/CARB, and ROT-treated zebrafish (e–h). Independently of the aggressive-like reactions and fluctuations among the testing day(s) and groups, ROT disrupted the social behavior, while VPA promoted a specific typology in contrast with LEV/CARB.

## 1. Introduction

The neurotoxic potential of MPTP (1-methyl-4-phenyl-1,2,3,6-tetrahydropyridine), 6-OHDA (6-hydroxydopamine), and paraquat (N, N′-dimethyl-4,4′-bipyridinium dichloride) as viable agents to generate PD-related symptoms is already well documented in the literature. Another compound that has gained increased interest with a toxicological profile and a broad spectrum of utility is ROT [1,2,3].

This plant-derived isoflavone is one of the oldest natural elements identified in several plants. The leaves, seeds, and stem of Mexican turnip (*Pachyrhizus erosus*), known under the trivial name of Jicama vine plants, and from roots of the *Fabaceae* family belonging to the genera *Derris*, *Lonchocarpus*, *Tephrosia*, and *Mundulea,* are specially processed to obtain ROT [4,5].

The *Lonchocarpus utilis* and *Nolina lindheimeriana,* native to South and North America, and *Lonchocarpus nicou* and *Derris elliptica* are also candidate species for obtaining ROT [4,5]. Due to its nature, the Federal Insecticide Fungicide Rodenticide Act [6] registered ROT in 1947.

ROT in small doses is safe if properly utilized, but it can be toxic to animals, fish, and humans. Compared to incomplete absorption by the gastrointestinal (GI) tract in fish, it is irrespective due to the absence of degrading enzymes in contrast to rodents [5].

Presently, ROT is confirmed to be a dopaminergic antagonistic that crosses the blood–brain barrier (BBB) and directly enters the central nervous system (CNS) and accumulates in cellular organelles, predominantly in the mitochondria, due to its lipophilic structure [1,2,3].

ROT induces dopamine neuronal toxicity [7], leading to a decline in adenosine triphosphate (ATP) generation and exacerbation in reactive oxygen species (ROS) via the inhibition of the complex I of the mitochondrial electron transport chain (ETC) [8]. Thus, ROT causes microglial activation, reflected by neuroinflammation [9], and aggregation of α-synuclein, known for their involvement in Lewy body pathology [10].

Fortunately, this field of research has received a lot of attention lately. However, little is known about ROT’s impacts on zebrafish behavior, particularly sociability and aggression. Based on these considerations, this study aims to evaluate the changes in the social component and level of aggression using the counterclockwise rotation parameter as a neurological pointer in a zebrafish (*Danio rerio*) chronically exposed to ROT for 32 days.

## 2. Materials and Methods

### 2.1. Animal Maintenance

We used forty adult (6–8 months), wild-type (WT), AB genetic line zebrafish (*Danio rerio*) purchased from an authorized local breeder from Iasi. The subjects were housed for 14 days in a 90 L dechlorinated water aquarium and for another 7 days in new 10 L tank(s). They were fed twice a day with TetraMin Flakes, while the water was changed daily in each experimental tank. The laboratory temperature was maintained at 26 ± 2 °C, pH 7.5, and 14 h light/10 h night cycle [11].

### 2.2. Ethical Note

Specimens were maintained and treated under the EU Commission Recommendation (2007), Directive 2010/63/EU of the European Parliament and of the Council of 22 September 2010 norms, referring to the guidelines for accommodation, care and protection of animals used for experimental and other scientific purposes. The implementation of this experiment was approved by the Ethics Committee of the Faculty of Biology, “Alexandru Ioan Cuza” University, Iasi, with the registration number 3936/26/11/2021.

### 2.3. Ant(Agonists) and Lactic Acid Lacteria Strains

ROT (5 g) was purchased from Toronto Research Chemicals, North York, Canada (# R700580), while VPA (100 g) from Sigma-Aldrich (#SLBC9758V), Saint Louis, MO, USA. LEV (250 mg) + CARB (25 mg) and PROBIO (3 g) that contained six *Lactobacillus* (*casei* W56, *acidophilus* W22, *paracasei* W20, *salivarius* W24, *lactis* W19, and *plantarum* W62) species and three *Bifidobacterium* (*lactis* W51 and W52, and *bifidum* W23) were bought from a local pharmacy. To avoid any conflicts of interest, the brand name of the product was kept under anonymity. ROT (2.5 µg/L) and VPA (0.5 mg/mL) were both dissolved in distilled water, whereas LEV + CARB (250 mg + 25 mg) and PROBIO (3 g) were dissolved and administered before the standard feeding routine for approximately half an hour to ensure the proper ingestion as unique doses using a 100 mL ratted balloon.

Our team [12] and Wang et al. [13] revealed that 2 µg/L over 21–28 days causes mild symptomatology. Thus, we performed some preliminary experiments prior to the actual protocol in which up to 5 zebrafish subjects per tank were exposed to three different doses (from 2 µg/L, 2.5 µg/L, and 5 µg/L) for 24 up to 72 h and concluded that 2.5 µg/L might be optimum, since 5 µg/L led to high mortality (data not shown) despite the existing evidence in the literature indicating a significant locomotor impairment (between 28 and 30 days of exposure) [14,15,16,17]. An analogous approach was applied for VPA, where we tested four doses (0.5 mg/mL, 2 mg/mL, 5 mg/mL, and 10 mg/mL). Amounts of 5 mg/mL and 10 mg/mL VPA led to high mortality, while in 2 mg/mL, they exhibited immobility episodes upon touching (data not shown). Based on these considerations, we managed to maintain the survival rate constant among subjects throughout the entire analyzed period.

### 2.4. Behavioral Testing

After acclimatization for 14 days, zebrafish (n = 5 per group) were randomly divided into eight groups, as follows: Group a was the CONTROL group, while Group b (0.5 mg/mL VPA), Group c (250 mg LEV and 25 mg CARB), Group d (3 g PROBIO), Group e (2.5 µg/L ROT), Group f (2.5 µg/L ROT in combination with 0.5 mg/mL VPA), Group g (2.5 µg/L ROT in combination with 250 mg LEV and 25 mg CARB), and Group h (2.5 µg/L ROT in combination with 3 g PROBIO) were the treated groups. The exposure solution was renewed daily in order to maintain a constant concentration. In addition, during the one-week pre-exposure period, the animals were transferred in vessels similar to the tests performed with the aim to become used to the stress of being caught and transferred as well with the novel configuration for observation. After the experimental accommodation, each experimental group was studied using the 2D and 3D approach over a 4 min period to set the baseline behavior, shown in our study as the initial behavior. No deaths were found in the control and treated groups after chronic exposure to chemicals.

#### 2.4.1. Anti-Social and Aggressivity Behaviors

The anti-social behavior and aggression tests were performed in a multipurpose cross maze closed by a transparent slit of Plexiglas and turned into a T-maze filled with system water (5 cm). We followed the standard protocol by placing the mirror and two social stimuli in the left arm. We focused on the tendency manifested to spend time in the central and right arm concerning the anti-social component and particularly the left arm for aggression. Each subject was left for half a minute for accommodation. The time length of each trial was 4 min per individual. Images were recorded with a professional infrared camera placed above the experimental chamber connected to a computer and analyzed using the software EthoVision XT 11.5, previously calibrated for these tests (Noldus Information Technology, Wageningen, The Netherlands) [12,18]. 

#### 2.4.2. Cycling Rotation

As already described by members of our group [19] and Kalueff [20], ‘*tight*’ cycling rotation but counterclockwise might indicate a high level of anxiety due to abnormal physiological response or selective drugs’ action, as in our case. In the counterclockwise rotations, the parameter of interest was analyzed by using the Track3D module of EthoVisionXT 14 video tracking software (Noldus Information Technology, Wageningen, The Netherlands). As above, each subject was left for half a minute to accommodate with the novel tank before starting the trial.

A schematic representation of the present study design can be found below (Figure 1).

### 2.5. Statistical Analyses

The normality and distribution were determined by Shapiro–Wilk test with Graph Pad Prism software (v 9.1.0.221, San Diego, CA, USA). Subsequently, multiple comparisons between the initial behavior and the days of testing within the groups were performed with one-way ANOVA followed by Dunnett’s test [21,22]. Trends were generated using OriginPro software (v 9.3-2016, OriginLab Corporation, Northampton, MA, USA).

## 3. Results

Although fluctuations in behavioral patterns are observable in all eight experimental groups, only in three did we observe a statistically significant difference over 32 days of analysis. We observed an abnormal pattern reflected by their circling tendencies in the (a) CONTROL group (D_24—*p* = 0.026) and (c) LEV/CARB group (D_24—*p* = 0.013) on the same day from a top view. Moreover, a significant difference was observed in the (d) PROBIO group (D_16—*p* = 0.022) from a side view perspective. Additional behavioral impairments in the remaining five groups were not observed (*p* > 0.05). However, in the non-exposed ROT groups (a–d), a constantly increasing pattern of rotation can be observed. In the remaining four groups (e–h) receiving ROT in combination with other agonists, this particular behavior was amplified without a significant difference (Figure 2).

Statistically significant differences were noted on separate days following the centralization and analysis of data on the time spent in the right and the central arm. Thus, group (b) supplemented only with VPA recorded a preference toward the right arm in D_1—*p* = 0.006 and D_8—*p* = 0.023, while group (c) who was given LEV/CARB, only in D_12—*p* = 0.008. Regarding group (e) ROT and group (f) ROT + VPA, zebrafish exhibited anti-social behavior in D_1—*p* = 0.002 and D_4—*p* = 0.012. The exploratory capacity was somewhat influenced, as the behavior corresponded to a state of anxiety in D_4—*p* = 0.005, D_20—*p* = 0.002, D_28—*p* = 0.004, D_32—*p* = 0.049 (b) VPA, and in D_4—*p* = 0.029, D_24—*p* = 0.021, D_28 and D_32—*p* < 0.001 (c) LEV/CARB. As already mentioned in the case of the other arm, the groups exposed to (f) ROT + VPA and (g) ROT + LEV/CARB were the only ones compared to (e) ROT and (h) ROT + PROBIO in which there were visible changes; D_12—*p* = 0.041, D_16—*p* = 0.005 (e) (ROT) and D_8—*p* = 0.008, D_24—*p* = 0.034 (f) ROT + VPA. What is intriguing is the lack of efficacy of lactic acid strains administered in the (d) PROBIO and (h) ROT + PROBIO groups but also in (g) ROT + LEV/CARB (*p* > 0.05). The (a) CONTROL group maintained a linear trend throughout the entire experiment, as in the (d) PROBIO group. Even though in (g) ROT + LEV/CARB and (h) ROT + PROBIO these fluctuations were much more visible, there were still no significant differences when comparing the initial behavior with each day of testing (Figure 3).

Compared to the pre-treatment period, even in the (a) CONTROL group, a deductible phenotype was observed based on the test performed. Interestingly, there was no statistically significant difference (*p* > 0.05) in the baseline behavior and the exposure time to left arm time. Including the right and central arm in (a) CONTROL but also in those that received (b) VPA, (c) LEV/CARB, (d) PROBIO, or in combination with (e–h) ROT, specific patterns of aggressive behavior were recorded (*p* < 0.05, 0.005, and 0.001) either in relation to the initial stage or between different days. However, the lack of significance should be noted in group (d) PROBIO (*p* > 0.05) at the time spent in the left arm but also by comparison with pre-treatment (*p* > 0.05) in the other two arms. It can be concluded that the PROBIO administered did indeed have a beneficial effect, an argument, which is not valid in the case of group (h) ROT + PROBIO (Figure 4).

## 4. Discussion

Zebrafish (*Danio rerio*) materialized as an optimal model to study a plethora of diseases [23]. It even outperformed rodent models, since their wide repertoire comprised normal and abnormal behaviors [24,25]. The social behavior might be attributed to their nature, living in shoals, being intrinsically collective creatures. They also portray well-documented expressions of fear and anxiety, and they can learn complex associations [26].

There are extensive data in the current literature describing the dose-time-dependent variable in inducing a PD-related phenotype in *Danio rerio.* Most of these studies, however, reflected the total distance swam, velocity, and freezing episodes reunited under the locomotion impairment umbrella rather than the social and aggressive components. Exposure to 2 µg/L ROT cause non-motor to mild symptoms [12,13], whereas 5 μg/L [14,15,16,17] up to 2 mg/L [27,28] might lead to excessive mortality, as in our case (unpublished data), or sufficient to induce a targeted phenotype.

VPA is nowadays an excellent stimulus for triggering symptoms that resembles autism spectrum disorder (ASD), demonstrating an inhibitory role following ROT exposure in rodents [29,30]. LEV/CARB are known to be dopaminergic agonists that, once ingested, cross the BBB in order to release dopamine, but in zebrafish, it seems to alleviate the cortisol level through the hypothalamic–pituitary–adrenal axis (HPA) [31]. Lastly, PROBIO proved to be the most powerful vehicle in restoring dysbacteriosis in fish, rodents, and humans [32].

Contrary to what we expected regarding the avoidance of chronic stress, we still observed a peculiar phenotype in the (a) CONTROL group and (c) LEV/CARB on the same day (*p* < 0.05). There was another instance when we noted a significant difference in behavior by comparison with the initial reference (*p* < 0.05) in the (d) PROBIO group. Notable phenotypical changes were absent in the counterclockwise rotation parameter in the groups exposed to ROT alone or a mixture, but relevant evidence occurred following the examination of anti-social behavior (Figure 2).

Groups (a) CONTROL, (d) PROBIO, (g) ROT + LEV/CARB, and (h) ROT + PROBIO did not register significant abnormal oscillations in behavior. Groups (b) VPA, (c) LEV/CARB, (e) ROT, and (f) ROT + VPA exhibited the most pronounced atypical behaviors with the most time spent in all three arms (*p* < 0.05, 0.005, 0.0005) (Figure 3). Afterward, we moved to evaluate the aggressivity level. The (a) CONTROL and (d) PROBIO groups exhibited a less pronounced level of aggressivity, comparable with the fluctuations displayed by the (b) VPA, (c) LEV/CARB, (e) ROT, (f) ROT + VPA, (g) ROT + LEV/CARB and (h) ROT + PROBIO groups (*p* < 0.05, 0.005, 0.001) (Figure 4).

It is noteworthy that we were not able to identify other teams whose purpose was to evaluate the harmful effect of ROT administration on the social and aggressive components. Considering that VPA is well known to induce symptoms that resemble the ASD, it was demonstrated on three distinct occasions that VPA may promote anxiety and hyperactivity, depending on the dose and exposure period.

Robea et al. [33] recently conducted a study on larvae zebrafish 6 days post-fertilization (dpf), aiming to expose them to 48 µM VPA for 24, 48, and 72 h. The group exposed to VPA for 72 h spent most of the time next to the mirror. There is also some controversy regarding this topic because Zimmermann et al. [34] contradict these findings, also using 48 µM. VPA influences the social component, anxiety, and locomotion rather than aggressive behavior. We highlighted the absence of any indicator pointing to a neurological disruption. This state was complementary to social behavior but correlated with high aggression. Liu et al. [35] brought solid evidence concerning how 20/100 μM VPA 7 h for 6 consecutive days caused social preference deficits in 24 h pf larvae, whereas acute exposure impaired locomotor activity. Neither intervention changed the behavioral response to light nor anxiety, considering that chronic exposure did not alter the locomotor activity.

There have been limited attempts to test LEV/CARB as triggers of aggressivity. Tan et al. [36] conducted a randomized controlled trial (RCT) in children diagnosed with Angelman Syndrome (AS) prophylactically treated with LEV. Per questionnaires applied and following the administration of 10 up to 15 mg/kg/day LEV, the cumulative data refute this possibility. However, an increase in dopamine (DA) level might exacerbate AS symptoms in a mouse model according to Riday et al. [37].

One possible explanation for the associated changes in human mood resides within the side effects. More specifically, the abrupt withdrawal or dose reduction in LEV increases the risk of neuroleptic malignant syndrome (NMS). NMS is also known as parkinsonism hyperpyrexia syndrome, which covers abnormal body temperature disturbance, spontaneous actions, and muscle rigidity. Patients might develop a dependence on LEV, which further explains the aggressive behavior [38].

Kutcher et al. [39] report marked interchanges, particularly in the aggressive-like reactions and submissive postures in LEV/CARB-exposed rats at 300 mg/kg subjected to intermittent semi-compulsory alcoholization and the joint kinetics of LEV/CARB. One method targeting the antioxidant balance stands in the use of antioxidants in L-DOPA mice as suggested by Hira et al. [40].

A regime based on lactic acid bacteria proved to promote improvements in the overall condition, but the interest congruent with our aim is lacking. Even though *Bifidobacterium longum* BB536 and *Lactobacillus rhamnosus* did not play a major role in the sociability of zebrafish exposed to 2 μg/L for 21 days [12], *Lactobacillus plantarum* and *rhamnosus* CECT8361/IMC 501 and *Bifidobacterium longum* CECT7347 are sufficient to alleviate anxiety-related behavior in larvae and adults [41,42,43].

## 5. Conclusions

In our studied animals, we observed an association in behavior in animals supplemented with ROT alone or a mixture and possible agonists. In this manuscript, the pre-determined doses administered in zebrafish (*Danio rerio*) for 32 days were enough to cause social deficits coupled with elevated moods of aggression. PROBIO exerted a beneficial effect on both analyzed parameters, diminishing aggressive-like symptoms. There were also circumstances where (a) CONTROL also manifested an impaired behavior but comparably attenuated by comparison with the remaining experimental groups. Due to the scarcity of data in the current literature and without knowing what the outcome might be, we are limited to behavioral studies that could constitute the first phase of a possible branch of research, also based on the reliance on multiple animals. We consider this manuscript to be the first launching pad for analyses that aim to elucidate both aggressive and social-related dysfunctionalities, even translated to clinical practice for PD patients. As can be concluded, this approach benefits from substantial potential, since immunohistochemistry coupled with analyses showing neuroinflammation and subsequent impairment of the enzymes responsible for the antioxidant status could offer further insight.

## Figures and Tables

**Figure 1 brainsci-12-00898-f001:**
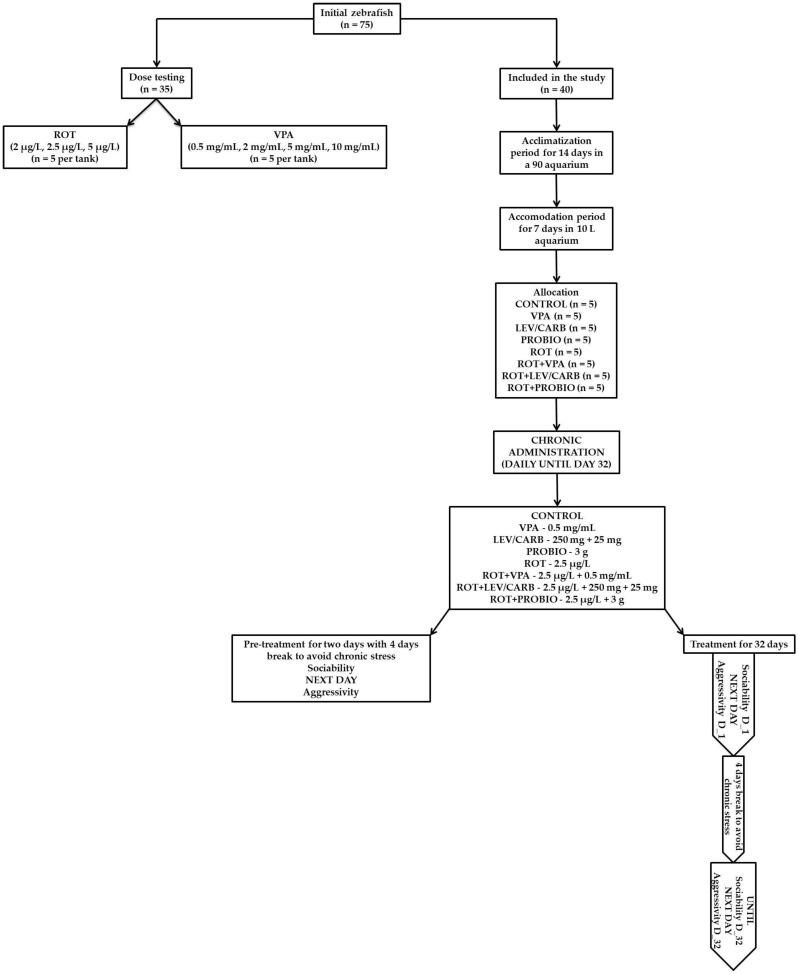
A CONSORT-style flow diagram of the study design.

**Figure 2 brainsci-12-00898-f002:**
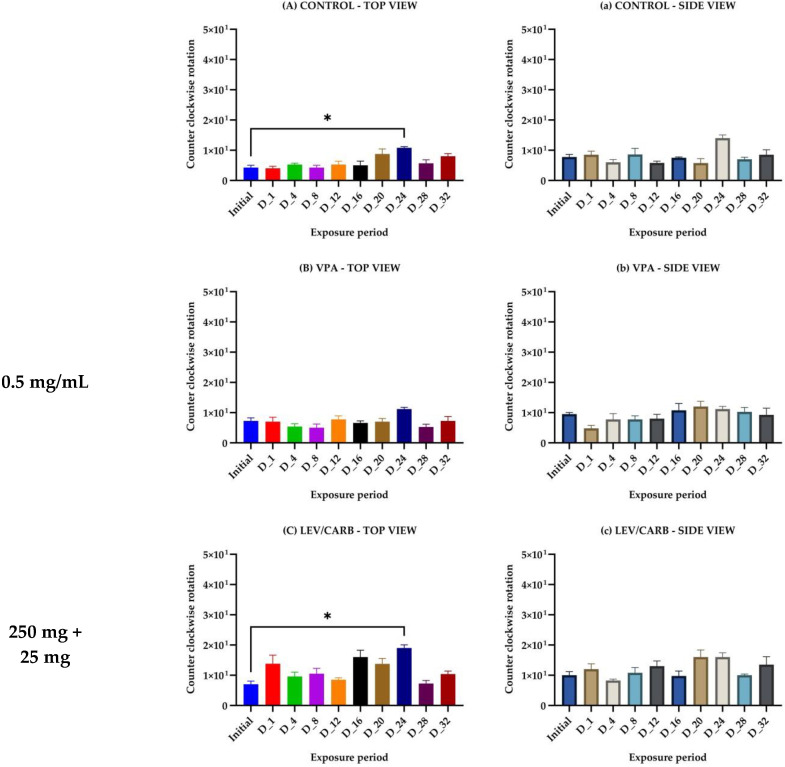
Counterclockwise rotation parameter in *Danio rerio* (n = 5) studied groups (values expressed as mean with SEM followed by Dunnett’s test; * *p* < 0.05).

**Figure 3 brainsci-12-00898-f003:**
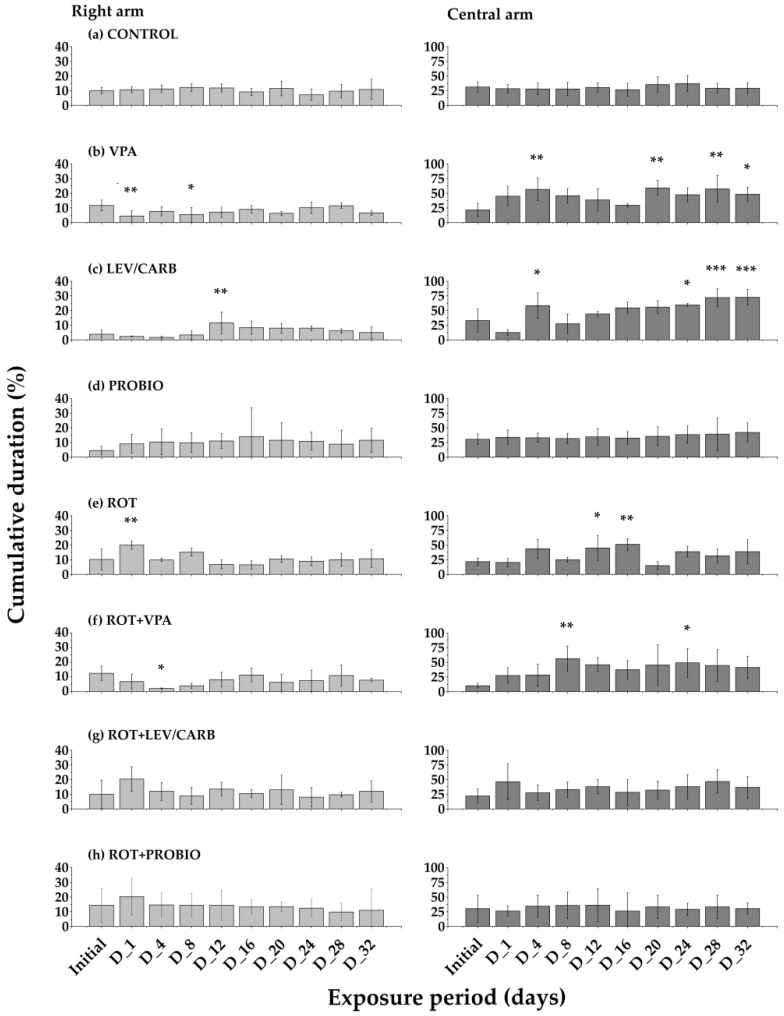
Anti-social pattern in *Danio rerio* (n = 5) studied groups and their tendencies toward both arms (values expressed as mean with SEM followed by Dunnett’s test; * *p* < 0.05, ** *p* < 0.005, *** *p* < 0.0005).

**Figure 4 brainsci-12-00898-f004:**
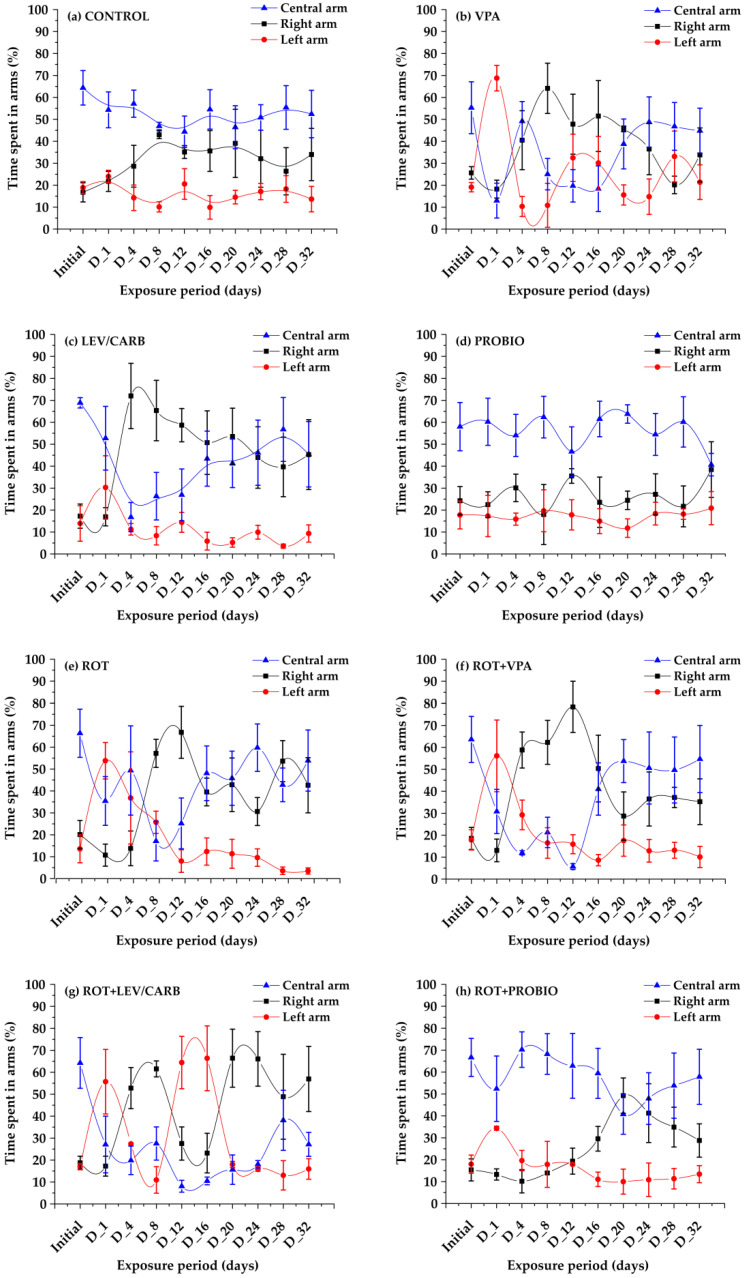
Aggressive-like patterns in *Danio rerio* (n = 5) studied groups and their tendencies toward all three arms (values expressed as mean with SEM followed by Dunnett’s test).

## Data Availability

The datasets used and analyzed in this study are available from the corresponding author on reasonable request.

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
