# Peer review of "Assessing Anti-Social and Aggressive Behavior in a Zebrafish (Danio rerio) Model of Parkinson’s Disease Chronically Exposed to Rotenone"

_brainsci, 2022, doi:10.3390/brainsci12070898_

Round 1
Reviewer 1 Report
Your manuscript describing the effects of rotenone and other therapeutically-approved compounds on zebrafish behavior provides interesting details about the nature of the rotenone zebrafish model.
Please provide a citation to support the concentration of rotenone used in your study. You appear to have used a 33 micro-Molar solution. Studies indicate an LD50 of 50 nano-Molar (https://www.frontiersin.org/articles/10.3389/fphar.2022.835827/full) but studies do vary. The rationale for your selected dose and the duration of exposure should be justified.
Please provide a CONSORT-style flow diagram depicting the timeline of the study. It is difficult to understand the timing of testing and outcome measurement, the duration of exposure to the different substances, the number of animals in each tank and the number of tank replications of each experiment, and other aspects of the study design. Figure 4 is insufficient because it does not convey these necessary details. I recommend replacing Figure 4, which is a simple overview of the steps of a scientific study.
Please indicate the number of animals that died during the course of the study.
Please describe how missing data were handled in your analysis. Please indicate when data were missing, such as by providing the n-value for the groups in each comparison.
Please describe when different post-hoc tests were used (Tukey HSD and Dunnett's). Why were different post-hoc tests used in different comparisons?
Because you have performed your study using repeated measures, it seems that the ANOVA is not an appropriate analysis. Instead, please consider using the Repeated Measures ANOVA as a robust and appropriate analysis. Identify the statistical test used for each analysis within your Results section and Figure legends so this information is clear and the reader can fully evaluate your conclusion.
Line 142, it is unclear how the CONTROL group is considered to be have abnormal rotations. To which group is this being compared? Was a within-group repeated measures test used for this analysis? Are you performing univariate analysis against an assumption that rotation will be evenly split 50:50 directions, or some other assumption? If the control and LEV/CARB groups are different from all other groups, perhaps the appropriate interpretation is that the other groups are abnormal, is this true?
Please clarify the pane identifiers in all of your figures. Each pane within the figure should have a unique identifier. For example, there should not be two panes identified as "(a)".
Please include a title for each graph in each figure. Please include x- and y-axis titles for each graph in each figure.
Please provide a legend for the figures that describes the data fully, such as the statistical analysis performed, the total number of replications per data point, the identity of the central tendency and variability (such as mean and standard deviation), the meaning of any symbols such as asterisks ("*" or "**"), and any other aspects necessary to interpret the data.
Please describe the limitations of your study. These can include a lack of replications or the reliance on multiple animals within the same tank which may result in a paired analysis that can magnify statistical artifacts. As an example, is it possible that the difference observed in CONTROLS was the result of an impurity introduced at a certain time, and the animals recovered shortly afterward?
I cannot agree with your conclusion because it uses the very strong word "proved". Instead, please use a cautious and scientifically appropriate, direct interpretation of your data from the studied animals. As an example, a conclusion should state, "In our studied animals, we observed an association in behavior in animals treated with a compound." Then, describe study designs that are necessary to overcome your limitations and expand your findings. You may also consider describing the potential implications or usefulness of your findings.
Author Response
Dear Reviewer #1,
Please consult the attached Response Letter regarding your comments
Kind regards and all the best,
Ovidiu-Dumitru Ilie

Reviewer 2 Report
The work presented by the authors arouses great reader interest with its relevance and the tasks set in the article. In the Introduction section, the history of the issue is briefly but meaningfully presented, the necessary literature references are given, and the main tasks of the work are formulated. The authors substantiate the effect of rotenone on the processes of neuroinflammation and activation of microglia at the biochemical and cellular level. However, the article contains mainly behavioral data, which are not supported by the results of IHC studies, which should have been presented in this paper as evidence for the inclusion of rotenone in the metabolism of zebrafish brain cells. The article presents the data of behavioral tests, however, the form of their presentation on the graphs (Fig. 1 and 2) is not very convincing. So, for example, for Fig. 1, 2 and 3 there are no labels along the y-axis; in fig. 2, there are no captions deciphering the values ​​along the x-axis. Captions for fig. 2 and 3 are too generalized, it is necessary to analyze the data presented in fig. 2 and 3 and make comments in the caption. Overall, the results section needs to be rewritten to provide clearer and more complete explanations for the behavioral observations presented. It is better to present the names of the experimental groups on graphs so that the reader can immediately understand what type of exposure was applied in each specific case. Without this, the presented figures look very uninformative. In the Discussion section, the authors discuss the results obtained, but do not indicate in which figure the discussed data are shown (Lin. 205-211). It is necessary to make changes in Figures 1, 2, 3 giving decodings to them and indicate the names of the experimental groups on the graphs.
Author Response
Dear Reviewer #2,
Please consult the attached Response Letter regarding your comments.
Kind regards and all the best,
Ovidiu-Dumitru Ilie

Round 2
Reviewer 1 Report
Authors:
Thank you very much for your extensive revisions. I now find your manuscript to be thoroughly described and more easily interpreted. I wish you best of luck on your future research.
Kind regards,
James
Reviewer 2 Report
Authors have significantly improved the text and made all the necessary corrections, the article can be published